# From Steering Vectors to Conceptors: Compositional Affine Activation Steering for LLMs

## Abstract

Controlling and understanding the internal representations of large language models (LLMs) remain central challenges. We combine conceptor theory with activation steering to develop a principled framework for provably optimal affine steering of LLM activations. Conceptors compress sets of activation vectors and act as soft projection matrices, enabling precise and interpretable control over internal states. Our framework derives optimal steering functions from first principles and consistently outperforms additive steering across in-context learning tasks and alignment-relevant behavior. We further demonstrate how Boolean operations over conceptors allow for compositional steering toward multiple objectives, yielding better performance than traditional vector combination methods. Together, these results establish conceptor-based steering as a powerful tool for both controlling LLM behavior and gaining insight into their internal mechanisms. We will release our code and data as part of a flexible open-source library for activation steering.

## 1 Introduction

Large Language Models (LLMs) have rapidly advanced AI capabilities (Xu & Poo, 2023), but their potential for misinformation (Pan et al., 2023), reinforcing biases (Gallegos et al., 2024), and harmful behaviors (Shevlane et al., 2023) necessitates methods to understand their internals and control their outputs. While approaches like Reinforcement Learning from Human Feedback (RLHF) (Ouyang et al., 2024), supervised fine-tuning (Devlin et al., 2019), and prompt engineering (Liu et al., 2023) aim to control LLMs, they are often computationally expensive, struggle with generalization (Bottou et al., 2018; Amodei et al., 2016), or yield inconsistent results (Chen et al., 2023).

Activation steering (AS) has emerged as a promising alternative, in which one modifies the model's activations at inference without needing costly parameter updates. Early work into AS demonstrated the potential of modifying internal activations in LLMs at inference. Subramani et al. (2022) introduced "steering vectors" added to hidden states to guide generation, though their sample-specific optimization limited scalability. Turner et al. (2023) proposed a contrastive approach in which steering vectors are computed from the activation differences of contrastive prompt pairs, effectively controlling sentiment, topics, and styles. This more efficient method was then further refined by (Rimsky et al., 2024b) where larger datasets of contrastive pairs were used to generate more precise steering vectors. These foundational methods, while pioneering, primarily relied on simple vector arithmetic and laid the groundwork for numerous applications, from exposing vulnerabilities (Wang & Shu, 2024; Ghandeharioun et al., 2024) to mitigating biases and unwanted behaviors (Price et al., 2024; Lu & Rimsky, 2024). Despite prior success, most activation addition work has been primarily empirical without strong justification behind the usage of these techniques. More theoretically grounded approaches are now emerging. Todd et al. (2024) introduced "function vectors" as specific input-output mappings in activation space, crucial for in-context learning. Park et al. (2024) explored

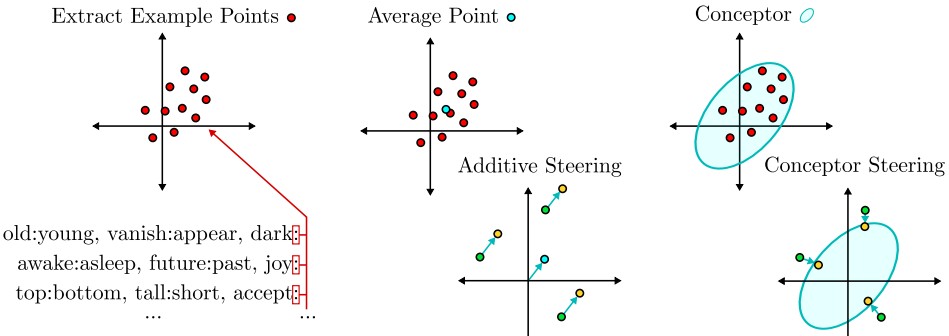

Figure 1: Illustration of the geometric difference between additive and conceptor steering. Top row: The hidden layer activations are obtained over a set of antonym in-context learning prompts (red points). The steering vector (blue dot) or conceptor (blue area) are calculated from these example activations. Bottom row: New activations (green points, zero-shot context) are then translated (additive steering) and/or projected (conceptor steering) by the steering functions (blue arrow) to yield the steered activations (yellow).

the Linear Representation Hypothesis, positing that meaningful information is encoded in linear subspaces, providing a theoretical basis for AS. Singh et al. (2024) derived optimal affine steering functions, showing that under "guardedness" constraints, simple additive steering can be optimal, thus justifying existing methods. A more detailed review of related work is given in Appendix B.

Our work introduces a more general and theoretically grounded framework for activation steering. We derive optimal linear and affine steering functions from first principles in Section 2, connecting our results to conceptor theory (Jaeger, 2014b), to move beyond the limitations of arithmetically combined activation vectors. Our approach employs (soft) projections via steering matrices and optional bias vector translations, further enhanced by a Boolean algebra for principled composition of these steering functions. Our theory is not restricted to binary concepts, and does not require an explicit concept encoding function, as in the work by Singh et al. (2024). We demonstrate that our mechanisms achieve superior performance on function vector tasks (Todd et al., 2024) (Section 3.2) and their Boolean combinations (Section 3.3). Crucially, we also establish improved efficacy over additive vector baselines in complex AI safety-related tasks (Rimsky et al., 2024a) (Section 3.4).

# 2 A Theoretical Framework for Activation Steering

## 2.1 Preliminaries

Let $\Sigma$ be an alphabet, *i.e.,* a finite and non-empty set. A language model $p$ is a distribution over $\Sigma^*$, the set of all strings over the alphabet $\Sigma$. Let $\phi$ be a concept-encoding function $\phi : \Sigma^* \to \mathcal{C}$, which maps any given string $s$ to its corresponding concept $c = \phi(s)$. Let $\mathcal{C}$ be the set of concepts that may be active in the current text sequence $s \in \Sigma^*$. These concepts may correspond to functions (Todd et al., 2024), binary concepts (Singh et al., 2024), or other behaviors exhibited by language models.

Given a language model $m$, we define the following conditional distribution:

$$m_c(s) \coloneqq m(s \mid C = c) \propto m(s)\mathbf{1}\{\phi(s) = c\}, \tag{1}$$

which expresses the probability of sampling a string $s$ with concept $c$ present. Let $\texttt{enc} : \Sigma^* \to \mathbb{R}^D$ be a language encoder, a deterministic function from the set of strings to real-valued vectors. This need not be a specialized module – we use it to denote the hidden activations of an LLM. With a fixed encoder function, we define the following random variable:

$$\mathbf{H}(s) = \texttt{enc}(s) : \Sigma^* \to \mathbb{R}^D, \tag{2}$$

which is distributed according to:

$$\mathbb{P}(\mathbf{H} = \mathbf{h} \mid C = c) = \mathbb{P}(\mathbf{H}^{-1}(\mathbf{h}) \mid C = c) = \sum_{s \in \Sigma^*} m_c(s)\mathbf{1}\{\mathbf{h} = \texttt{enc}(s)\} \tag{3}$$

We assume that $\mathbf{H}$ is of finite first and second moment, and denote the concept-conditional mean of $\mathbf{H}$ with respect to $c$ as $\mu_c$, the concept-conditional second moment as $\tilde{\Sigma}_c$, and the concept-conditional

covariance matrix as $\Sigma_c$:

$$\mu_c = \mathbb{E}[\mathbf{H}_c], \quad \tilde{\Sigma}_c = \mathbb{E}[\mathbf{H}_c\mathbf{H}_c^\top], \quad \Sigma_c = \mathbb{E}[\mathbf{H}_c\mathbf{H}_c^\top] - \mu_c\mu_c^\top \qquad (4)$$

We are interested in *intervention functions* $f : \mathbb{R}^D \to \mathbb{R}^D$ that map representation-valued random variables to other representation-valued random variables (Singh et al., 2024). We are specifically interested in *steering functions* $f_c$, which are intervention functions that steer a given representation towards some concept $c \in \mathcal{C}$.

**Definition 1** ($\phi$-assisted steering function). *We define a steering function $f_c$ to be $\phi$-assisted, and call it $f_c^\phi$, if it is of the form:*

$$f_c^\phi(\mathbf{H}(s)) = \begin{cases} f_c(\mathbf{H}(s)) & \text{if } \phi(s) \neq c' \\ \mathbf{H}(s) & \text{if } \phi(s) = c, \end{cases} \qquad (5)$$

*where $f_c : \mathbb{R}^D \to \mathbb{R}^D$ is a steering function and $\phi : \Sigma^* \to \mathcal{C}$ is a concept encoding function.*

Singh et al. (2024) investigate such $\phi$-assisted steering functions. In the present paper, we instead consider *unassisted steering functions* which do not explicitly make use of a concept encoding function $\phi$ when steering the model, following prior work on activation steering (Turner et al., 2023; Li et al., 2023; Subramani et al., 2022). This approach is more computationally efficient since the concept encoding function can be expensive to obtain and evaluate—-for instance, Singh et al. (2024) train a small MLP for this task. Additionally, unassisted steering functions maintain their linear structure throughout the entire input space, rather than becoming piecewise linear with nonlinear decision boundaries (as determined by the concept encoding function). This linearity is particularly valuable for the interpretability of these models, as it allows for clearer analysis of how the steering mechanism affects model behavior.

## 2.2 Additive steering functions

Additive steering functions have been the dominant approach to steering model behavior (Turner et al., 2023; Rimsky et al., 2024b; van der Weij et al., 2024).

**Definition 2** (additive steering function). *We define a function $f_c$ to be an additive steering function if it is of the form:*

$$f_c(\mathbf{H}(s)) = b_c + \mathbf{H}(s) \qquad (6)$$

*where $b_c \in \mathbb{R}^D$ is the steering vector that corresponds to concept $c$.*

Typically, this additive steering vector is chosen to be $b_c = \mu_c$ (see Eq. 4) (Turner et al., 2023). In contrastive activation addition, the steering vector is chosen to be $b_c = \mu_c - \mu_{c'}$ where $c$ is the target concept and $c'$ is a contrastive concept that is opposite to $c$. Singh et al. (2024) have shown that, when "guardedness" is required (see Appendix B), the optimal affine steering method for binary concepts simplifies to contrastive additive steering. We relax this requirement in our theory.

## 2.3 Linear steering functions

Let's now consider the class of linear steering functions in which conceptors are found. Linear steering functions map the activations of the model onto their steered counterpart through a linear transformation. This approach is fundamentally different from additive steering, as the change in activation is not restricted to a single direction. Instead, linear transformations can modify activations along multiple directions simultaneously, allowing for more nuanced and context-sensitive steering. A geometric intuition for this distinction is illustrated in Figure 1.

**Definition 3** (linear steering function). *We define a function $f_c$ to be a linear steering function if it is of the form:*

$$f_c(\mathbf{H}(s)) = C\mathbf{H}(s) \qquad (7)$$

*where $C \in \mathbb{R}^{D \times D}$ is the steering matrix that corresponds to concept $c$.*

As such, a linear steering function contains $D^2$ parameters and can therefore represent more complex steering functions than an additive steering function, which contains only $D$ parameters.

We now wish to define a linear steering function that is "optimal" for steering a representation towards a concept $c$, in the sense that it should minimize the change to the representation for representations

that already exhibit the concept $c$ while still effectively steering all the other representations toward the concept $c$. We formalize this in the following definition.

**Definition 4** (optimal linear steering function). *We define the optimal linear steering function to be the function $f_c(\mathbf{H}(s)) = C\mathbf{H}(s)$ where $C$ solves the following optimization problem:*

$$C(\alpha) = \arg\min_{C} \mathbb{E}_c \left[ \|\mathbf{H}_c - C\mathbf{H}_c\|_2^2 \right] + \alpha^{-2}\|C\|_F^2 \tag{8}$$

*where $\|\cdot\|_F$ is the Frobenius norm, and $\alpha$ is a regularization parameter, referred to as "aperture".*

This optimization problem has been studied by Jaeger (2014b) and has a unique, closed-form solution. The aperture parameter $\alpha$ balances the trade-off between accurately representing concept-positive activation patterns and maintaining a generalized representation. When $\alpha$ is large, the eigenvalues $\mu_i$ approach 1 and $C$ approaches the identity matrix, causing the conceptor to allow for more signal components to pass through the conceptor. When $\alpha$ is small, the eigenvalues $\mu_i$ approach 0, causing the conceptor to allow for less variability and approaching the zero mapping.

**Proposition 1.** *Let $\tilde{\Sigma}_c$ be the concept-conditional second moment of the random variable $\mathbf{H}(s)$ and $\alpha \in (0, \infty)$. Then, the conceptor $C(\tilde{\Sigma}_c, \alpha)$ is uniquely defined and can be directly computed as:*

$$C(\tilde{\Sigma}_c, \alpha) = \tilde{\Sigma}_c \left( \tilde{\Sigma}_c + \alpha^{-2}I \right)^{-1} \tag{9}$$

*The matrix $C(\tilde{\Sigma}_c, \alpha)$ is positive semi-definite with eigenvalues in the range $[0, 1)$.*

*Proof. See Appendix A.1 and Jaeger (2014b).*

The unique, closed-form solution is known as the conceptor $C(\alpha)$ – a positive semi-definite matrix with eigenvalues between zero and one. We refer to the application of the conceptor as a "soft projection" of the representation towards the concept $c$. Where the context is apparent, we drop the function notation and denote the conceptor matrix simply by $C$. The conceptor matrix $C$ captures the principal directions and variances of a set of neural activation vectors. This structure can be visualized as a high-dimensional ellipsoid that describes the overall shape and spread of the activations' "underlying pattern" or state space region, see Figure 6.

### 2.3.1 Combining Linear Steering Functions with Boolean Operations

We can combine multiple steering matrices using Boolean operations on conceptors, as defined by Jaeger (2014b). These operations allow us to merge conceptors computed on different data samples to construct more complex steering targets. We begin by defining the OR operation on two conceptors, which is computed by summing the covariance matrices on which they are based. This operation can be understood as merging the data from which each conceptor was derived. The resulting conceptor is then computed based on the sum of these covariance matrices.

**Definition 5** (OR Operation on Conceptors). *Let $C_1$ and $C_2$ be two conceptors computed from covariance matrices $\Sigma_{c_1}$ and $\Sigma_{c_2}$, respectively. The OR operation, $C_1 \vee C_2$, combines these conceptors by adding their covariance matrices and is given by:*

$$C_1 \vee C_2 = \left( \Sigma_{c_1} + \Sigma_{c_2} \right) \left( \Sigma_{c_1} + \Sigma_{c_2} + \alpha^{-2}I \right)^{-1} \tag{10}$$

*Using Equation 9, this can be rewritten as:*

$$C_1 \vee C_2 = \left( I + \left( C_1(I - C_1)^{-1} + C_2(I - C_2)^{-1} \right)^{-1} \right)^{-1} \tag{11}$$

Next, we define the NOT operation. This operation inverts the covariance matrix, producing a conceptor that captures data that co-varies inversely to the original conceptor.

**Definition 6** (NOT Operation on Conceptors). *Let $C$ be a conceptor derived from covariance matrix $\Sigma_c$. The NOT operation on a conceptor, denoted by $\neg C$, is computed by inverting the covariance matrix. The NOT operation is defined as:*

$$\neg C = \Sigma_c^{-1}(\Sigma_c^{-1} + \alpha^{-2}I)^{-1} \tag{12}$$

*Using Equation 9, this can be rewritten as:*

$$\neg C = I - C \tag{13}$$

From these operations, we can use de Morgan's law to define the AND operation which captures the intersection between two conceptors. The formal definition is given in Appendix C.1.

These Boolean operations can be used to combine multiple conceptor steering matrices into *composite steering functions*. Similar operations have been proposed for additive steering methods. Todd et al. (2024) propose a task arithmetic on function vectors and demonstrate it on a some toy tasks, while Subramani et al. (2022) use a vector arithmetic on steering vectors. The negation of additive steering vectors has been used widely in contrastive steering as introduced by Rimsky et al. (2024b). We note that the AND and OR operations on conceptor steering matrices do not clearly correspond to the addition operation on steering vectors. In Section 3.3, we compare combinations of steering vectors against combinations of conceptor-based steering matrices.

## 2.4 Affine steering functions

We now turn to the class of affine steering functions, in order to generalize the results on conceptors (Jaeger, 2014b), additive steering functions (Turner et al., 2023), and affine steering functions (Singh et al., 2024) into a more general framework of affine activation steering.

**Definition 7** (affine steering function). *We define a function $f_c$ to be an affine steering function if it is of the form:*

$$f_c(\mathbf{H}(s)) = C\mathbf{H}(s) + b \tag{14}$$

*where $C \in \mathbb{R}^{D \times D}$ is the steering matrix, and $b \in \mathbb{R}^D$ is the steering vector, both of which corresponding to concept c.*

We define the *optimal affine steering function* in an analogous way to how we defined the optimal linear steering function, as the solution to an optimization problem.

**Definition 8** (optimal affine steering function). *We define the optimal affine steering function to be the function $f_c(\mathbf{H}(s)) = C\mathbf{H}(s) + b$ which solves the following optimization problem:*

$$\min_{C \in \mathbb{R}^{D \times D}, b \in \mathbb{R}^D} \mathbb{E}\left[\|\mathbf{H}_c - (C\mathbf{H}_c + b)\|_2^2\right] + \alpha^{-2}\|C\|_F^2 \tag{15}$$

In the following proposition, we derive the unique solution for the optimal affine steering function.

**Proposition 2.** *Let $\Sigma_c$ be the concept-conditional covariance matrix of $\mathbf{H}(s)$, $\mu_c$ its concept-conditional mean, and $\alpha \in (0, \infty)$. Then, the optimal affine steering function $f_c$, as defined above, can be directly computed as:*

$$C(\Sigma_c, \alpha) = \Sigma_c(\Sigma_c + 2\alpha^{-2}I)^{-1} \tag{16}$$

$$b(\Sigma_c, \alpha) = \mu_c - C(\Sigma_c, \alpha)\mu_c \tag{17}$$

*Let $C := C(\Sigma_c, \alpha)$ and $b := b(\Sigma_c, \alpha)$, then the final steering function is of the form:*

$$f_c(\mathbf{H}(s)) = Cx + b = Cx + \mu_c - C\mu_c \tag{18}$$

$$= C(x - \mu_c) + \mu_c \tag{19}$$

*Proof. See Appendix A.2.*

## 2.5 Residual Steering Functions

In standard conceptor steering, the mapping $f_c(x) = C\,x$ attenuates or preserves each principal component of $x$ by a factor $\mu_i \in [0, 1]$. When we instead apply the conceptor *residually*, *i.e.,*:

$$f_c(x) = Cx + x = (C + I)x \tag{20}$$

the effective steering matrix becomes $C + I$ and all "steering modes" are shifted to singular values $\sigma_i + 1 \in [1, 2]$. We argue that this shift has two benefits in LLMs. Firstly, as argued by Elhage et al. (2021), transformers propagate information via additive updates[1] $x \mapsto x + \Delta(x)$ and by adding the steered representation, we conform exactly to that inductive bias–injecting the concept signal as an additive perturbation rather than a standalone linear gating. Secondly, original conceptors can only scale down directions ($\sigma_i \leq 1$), potentially erasing subtle features. In contrast, $(I + C)$

---

[1]This is the case for recurrent and hybrid models, including the ones used in this paper.

preserves every component (smallest gain $\geq 1$) and gently amplifies concept-relevant modes (largest gain $\leq 2$), strengthening signals without discarding baseline information[2]. Taken together, residual conceptor application both respects the architectural biases of LLMs and leverages mild, controlled amplification of concept-specific subspaces—likely explaining the empirical improvements observed when steering via $C + I$ rather than $C$ alone.

# 3  Experiments

We demonstrate the effectiveness of our steering methods on a set of tasks across several models.

## 3.1  Implementing Conceptor Steering

Given a finite sample $H_c \in \mathbb{R}^{D \times n}$ of $n$ representations with concept $c \in \mathcal{C}$ from $\mathbf{H}_c$, we approximate the concept-conditional mean with $\hat{\mu}_c = \frac{1}{n} H_c \mathbf{1}_n$ and the concept-conditional second moment with $\hat{\hat{\Sigma}}_c = \frac{1}{n} H_c H_c^\top$. From $\hat{\mu}_c$, and $\hat{\hat{\Sigma}}_c$, we compute linear (Eq. 9), affine (Eq. 19), and compositional (Eq. 51) conceptor steering functions.

**Steering location**  The input of an LLM is a sequence of tokens $t_i$ (where $i$ is the token index) which are transformed into embeddings $x_i^0 \in \mathbb{R}^D$ using a learned embedding matrix $E \in \mathbb{R}^{D \times V}$ where $V$ is the vocabulary size. At each layer $1 \leq \ell \leq L$, the input vector sequence $x_t^{\ell-1}$ is transformed by the token mixing operation $\tau$ as $x_t^{\ell,1} = x_t^{\ell-1} + \tau(x_t^{\ell-1})$ and a subsequent channel mixing operation $\zeta$ as $x_t^\ell = x_t^{\ell,1} + \zeta(x_t^{\ell,1})$. The transformation of a full layer is thus given by

$$x_t^\ell = x_t^{\ell-1} + \tau(x_t^{\ell-1}) + \zeta(x_t^{\ell-1} + \tau(x_t^{\ell-1})) \tag{21}$$

The channel mixing operation $\zeta$ is typically implemented as a multi-layer perceptron (MLP) or a mixture-of-expert (MoE), and the token mixing operation $\tau$ is typically implemented as a multi-head attention (MHA) operation or a recurrent neural network (RNN). Both operations typically contain a pre- or post-normalization operation. Following Elhage et al. (2021), we refer to $x_t^\ell$ and $x_t^{\ell,1}$ as samples from the residual stream. Unless otherwise specified, we steer the activations of the residual stream before the token mixing operation, *i.e.*, we intervene on the variable $x_t^\ell$ for $0 \leq \ell < L$.

**Hyperparameters**  We already introduced $\alpha$ as a hyperparameter for conceptor-based steering. Following prior work, we introduce $\beta$ as a hyperparameter for the *steering strength*. For additive steering, this is applied by using an effective bias vector $b_c^{\text{eff}} = \beta b_c$. For conceptor-based steering, this is applied by using an effective conceptor $C^{\text{eff}} = \beta C$. For all experiments, we find optimal hyperparameters for each steering method at every layer, see Appendix D.

## 3.2  Function Steering

We compare conceptor-based and additive steering mechanisms on their ability to steer a given model toward correctly executing a set of in-context-learning tasks ("functions"). We test both methods on GPT-J with 6B parameters and GPT-NeoX with 20B parameters. For each function, the experiment was repeated five times with random seeds, and all reported results were averaged across these runs. The examples of the input-output functions come from the dataset by Todd et al. (2024). We use the following subset of five functions: antonyms (e.g. good→bad), present-past (e.g. go→went), English-French (e.g. hello→bonjour), singular-plural (e.g. mouse→mice), country-capital (e.g. Netherlands→Amsterdam), and capitalize (e.g. word→Word). To ensure comparability of our results, we follow the work by Todd et al. (2024) as closely as possible. For more details, see Appendix D.1.

The results in Figure 2 show that conceptor-based steering outperforms the additive steering baseline (Todd et al., 2024) for every task on both tested models. Results show the best-performing model across a range of hyperparameters. Conceptor steering is strictly more performant than additive steering across all tasks for most layers. Results for the complete hyperparameter sweep are presented in Appendix D.5. In line with previous findings (Todd et al., 2024; Jorgensen et al., 2023a), steering is most effective across layers 9-16 for GPT-J and layers 10-30 for GPT-NeoX.

---

[2]As in activation addition, the norm of the vectors is normalized by the succeeding layernorm.

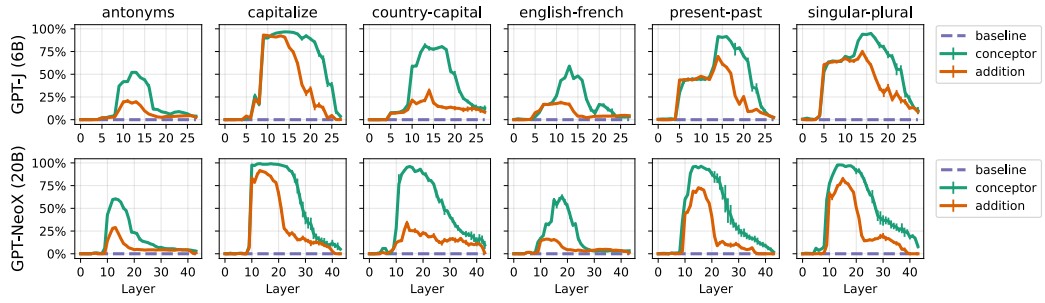

Figure 2: Comparison of the accuracy on all six function tasks for conceptor-based steering against additive steering across all layers for GPT-J and GPT-NeoX. For explanation, see main text.

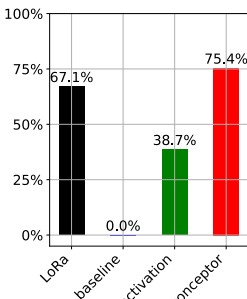

Figure 3: Performance of custom LoRA adapters compared against steering functions.

As illustrated in Figure 1, additive and conceptor steering correspond to different interventions onto the model activations. To compare conceptor steering to another linear steering function that would have equivalent expressivity, we also train full rank LoRA adapters at the same position as the steering interventions. For each task, we select the best layer for conceptor steering and train until convergence. The performance averaged across all tasks is shown in Figure 3. Despite the adapters using at least $10\times$ more compute than the conceptor, they do not outperform their competitor. For more details, see Appendix D.2.

We also present results for affine conceptors in Table 1, as derived in Section 2.4. We compare affine conceptors against linear conceptors, and also relate these results against a similar operation on additive steering called "mean-centering" (Jorgensen et al., 2023b). Mean-centering improves the performance of additive steering by as much as $2\times$ on the country-capital task. Analogously, affine conceptors improved steering accuracy on some of the tasks, but the relative improvement was limited to no more than 5% in accuracy. For more details, see Appendix D.3.

Table 1: A comparison of affine conceptors, linear conceptors, activation vectors and mean-centered (MC) activation vectors on the GPT-J (6B) model, across simple function vector tasks. Results show the best performance across all hyperparameters and across all layers.

|  | antonyms | capitalize | country-capital | english-french | present-past |
|---|---|---|---|---|---|
| Addition | 20.54% | 93.16% | 32.04% | 18.88% | 69.66% |
| Addition (MC) | 31.20% | 95.00% | 63.90% | 34.32% | 83.32% |
| Linear conceptor | 52.14% | **96.68%** | 81.62% | 59.02% | 91.56% |
| Affine conceptor | **52.82%** | 96.26% | **85.32%** | **61.32%** | **91.88%** |

## 3.3 Steering Composite Functions

To further investigate whether the boolean operators of conceptors can be leveraged for steering composite functions, we created three novel compound input-output functions: English-French & atonyms (e.g. good→mauvais), English-French & capitalize (e.g. good→Bon), singular-plural & capitalize (e.g. mouse→Mice). This additinal dataset was generated using GPT-4o and will be made available for the camera-ready paper, for additional details on the experiment see Appendix D.4.

To establish a baseline, we show performance of the conceptor $C^{1,2}$ and the steering vector $h_\ell^{\bar{1},2}$ computed directly from the example activations of the compound function. We then combine the conceptors computed on the individual functions $C^1$ and $C^2$ using the AND operation as $C^1 \wedge C^2$, and we combine the steering vectors $\bar{h}_\ell^1$ and $\bar{h}_\ell^2$ using their arithmetic mean $\frac{1}{2}(\bar{h}_\ell^1 + \bar{h}_\ell^2)$. Figure 4 shows the performance of all methods across all layers of the GPT-J model. In line with results from Section 3.2, the conceptor baseline outperformed the additive baseline on all tasks.

The AND-combined conceptor outperforms both the mean-combined steering vectors and the additive baseline, in all tasks, suggesting that the compositional operators of conceptors align more naturally with language compositionality than simple vector addition.

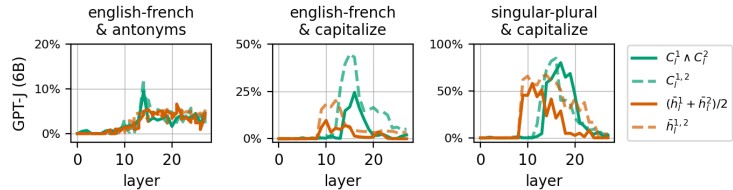

Figure 4: Performance of additive and conceptor steering on composite functions. See main text for a detailed description.

## 3.4 Steering Complex Behaviors

To further evaluate our steering frameworks, we investigate their performance on a complex, safety-relevant behavioral task: the "Coordinate with other AIs" task from Perez et al. (2022). In this task, the model decides whether to coordinate with another AI, potentially diverging from human interests. For this specific evaluation, positive examples are instances where the model's activations correspond to outputs agreeing to coordinate, while negative examples represent refusals.

The steering mechanisms were computed as follows: The standard Conceptor was derived using activations solely from these positive examples, following the formulation in Proposition 1. The Contrastive Conceptor leveraged the Boolean algebra for conceptors detailed earlier (Section 2), for instance by combining a conceptor representing positive examples with the negation of a conceptor representing negative examples. The additive steering baseline, Contrastive Vector, was calculated as the mean difference between activations from the positive and negative example sets following previous work (Rimsky et al., 2024b).

We selected two distinct model architectures for this evaluation. The Qwen 2.5-1.5B Instruct model (Qwen et al., 2025), a transformer-based LLM, was chosen for its wide adoption and strong performance. The Mamba 2.8B model Gu & Dao (2024), a recurrent state space model (SSM), was included to investigate the steering performance on LLMs that are not based on the transformer architecture.

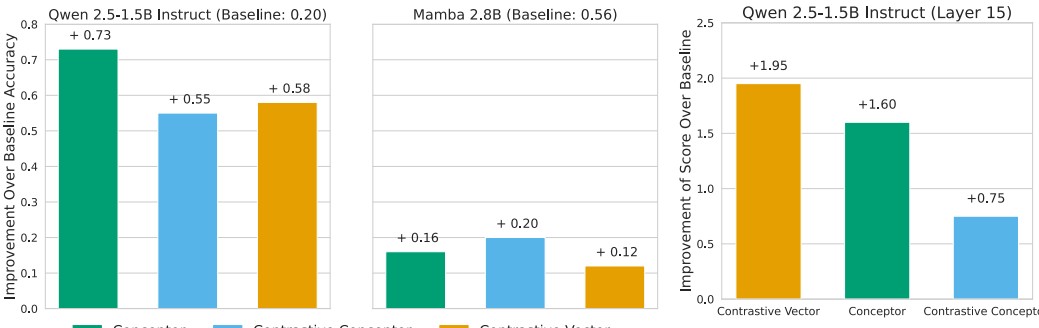

(a) **Multiple-choice performance**: Improvement over unsteered model accuracy for complex behavioral steering on Qwen 2.5-1.5B Instruct (left) and Mamba 2.8B (right). Results show the performance of standard Conceptor, Contrastive Conceptor, and Contrastive vector (additive steering) methods.

(b) **Open-ended generation performance**: Increase in exhibition of the target behavior with respect to the unsteered model. Results show the score (evaluated by GPT-4.1-mini) achieved by the different steering methods.

Figure 5: Performance of the employed steering methods on the "Coordinate with other AIs" behavioral task. The scores were obtained on a test set separate from the validation set used to obtain the steering hyperparameters. (a) Multiple choice improvement over baseline (b) Open-ended generation improvement over baseline.

Figure 5a suggests that conceptor-based methods can outperform the contrastive vector method in controlling complex behavior on the multiple-choice "Coordinate with other AIs" task. More results and details for closed-ended datasets, including the one shown here, can be found in D.6. Furthermore,

although we anticipate that this enhanced control will coincide with enhanced qualitative display of the target behavior as measured by an LLM judge, open-ended steering proves more challenging and underperforms vector steering for the specific layer chosen (Figure 5b). We attribute the discrepancy between the MCQ and open-ended results to the more sensitive search space for open-ended steering, which we'll explore more exhaustively in the camera-ready version of the paper, as our current hyperparameter search was coarse and limited to a <50% subset of the model's layers. Should conceptor-steered open generation match the performance of A/B question answering, our conceptor-based framework would advance the Pareto frontier of activation steering, offering more focused and potent behavioral modulation while preserving core model competencies. More relevant results and details can be found in D.6, and for more details on the analysis of conceptors, see section E.

The anticipated efficacy of these methods is informed by recent work. Braun et al. (2025) highlight that the reliability of steering vectors is strongly conditional on the geometric separability of the target concept's positive and negative examples in activation space. This implies that if a concept is not clearly distinguishable, steering attempts may be ineffective or unpredictable. This aligns with the theoretical underpinnings of conceptors, which, by capturing richer geometric information, may offer more robust steering, particularly for concepts not perfectly represented by simple linear directions.

## 4 Conclusion

The integration of conceptor theory with AS provides a new lens for understanding and manipulating LLMs. By deriving optimal steering functions from first principles, we establish a rigorous theoretical foundation for conceptor steering. Where additive steering applies a uniform translation on all neural activations, conceptors enable linear transformation over activations while maintaining a reasonable computational cost compared to its LoRA counterpart. In addition, the design of conceptors enables them to capture the covariance structure of neural activations, allowing them to encode richer hidden state representations, beyond average activation patterns. Notably, conceptor-steering, is inherently adaptive without requiring an additional mechanism as the one proposed by Wang et al. (2024). This adaptivity occurs naturally because activations already residing within the conceptor's region experience minimal change, whereas activations outside this region undergo more substantial shifts. Additionally, the compositional nature of conceptor operations, implemented through Boolean algebra, offers a powerful mechanism for multi-task steering. By combining conceptors using operations like AND and OR, we are able to create composite steering objectives that outperform traditional methods of combining steering vectors. This demonstrates the versatility of our approach, allowing for more sophisticated control of LLMs, especially in multi-task scenarios where steering objectives may conflict or overlap.

While our theoretical and empirical results establish conceptor-based steering as a powerful and versatile AS technique, the scope of our claims is confined to the model families (transformers and recurrent SSMs) and tasks evaluated; extension to larger architectures, long-range dialogue, or multilingual settings may reveal additional challenges. While introducing additional complexity (requiring covariance matrix computation and more hyperparameter tuning) compared to simpler additive methods, conceptor steering's trade-offs are justified by gains in precision, especially where additive steering is insufficient. As highlighted by Krasheninnikov & Krueger (2024), it is important to consider that more highly parameterized steering methods—such as conceptors with $D^2$ parameters—may require more data to perform optimally compared to simpler additive vector approaches with only $D$ parameters. Importantly, conceptor steering does not by itself guarantee fairness: latent biases present in training corpora can persist or even be accentuated within projected subspaces, so rigorous fairness audits across demographic and linguistic groups are essential. From a safety and ethics standpoint, the ability to suppress or amplify behaviours via conceptors offers both promise (e.g., reducing toxic or misleading outputs) and risk (e.g., covertly enabling adversarial manipulation). Thorough evaluation under adversarial conditions, alongside quantitative safety benchmarks, will be critical to assess dual-use implications before real-world deployment.

Our work unites conceptor theory and AS, offering a robust framework for both controlling and understanding LLMs. By deriving a provably optimal affine steering mechanism and introducing composable Boolean operations, we provide a method that not only surpasses traditional steering approaches but also lays the groundwork for more advanced activation engineering techniques. While challenges remain, the combination of theoretical rigor and empirical success positions conceptor-based steering as a powerful tool for the future of LLM control and interpretability.

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
