# OpenReview forum: "From Steering Vectors to Conceptors: Compositional Affine Activation Steering for LLMs"
_NeurIPS.cc/2025/Conference — Submitted to NeurIPS 2025_

### Official Review · Reviewer_mWE6 · 2025-07-01

**Clarity:** 2
**Significance:** 3
**Originality:** 3
**Rating:** 3
**Confidence:** 4

**Summary:**

In this paper, the authors proposed a conceptor-based affine transformation on LLMs' activations to steer the models' function or behavior. It primarily applied conceptor, which has been used in RNNs to specify patterns from state representations and drive the RNNs with these patterns, onto transformers' activations to achieve similar outcomes by projecting activations to the "ellipsoidal subspace" comprising example activations that represent a function or behavior with a minimal (in terms of Frobenius norm) matrix. The authors also made adaptions specifically for transformers, including combining the projection with conventional additive steering vectors, incorporating "mean-centering", applying conceptor residually, etc. While there are other existing activation steering methods that don't solely depend on a fixed steering vector, the authors joined them in claiming vanilla steering vectors are insufficient but didn't directly compete with any of them. The proposed steering method was tested extensively in steering a variety of moderately-sized LLMs to perform different ICL tasks zero-shot or in a natural language context against the vanilla additive steering vector and saw consistent advantage over it. The authors also put the logical composition hypothesis of conceptors to test and yielded positive results.

**Questions:**

**Q1**: The theory behind the proposed conceptor-based steering of transformers is a bit confusing, especially with the presence of steering vectors. I thought, albeit not explicitly mentioned by Jaeger, the states (or activations here) are supposed to be zero-mean in order for the conceptor to capture the covariance which is expected to encode a concept. However, since there do exist CAA-fashioned steering vectors that non-trivially promote the same concept, it indicates that the mean is away from origin by design. Then although the point clouds may still look like an ellipsoid as in Figure 1, the ellipsoid is not a subspace and the conceptor isn't necessarily "drawing" activations closer to it in the depicted way. In an extreme example in the same 2D world, if the example activations cluster around $v\pm\varepsilon\cdot v^\perp$ for some small $\varepsilon$ w.r.t. $|v|$, the conceptor would be dominated by the mean shift while ignoring the actual axis of variance, resulting in a projection onto $v$, so $-v$ which is supposed to exhibit the negative concept won't be affected by the conceptor. Figure 1 was used to explain the linear conceptor but it seems to be only proper if it is meant for the affine conceptor which relies on CAA-fashioned steering vectors to accound for the mean shift, and there is supposed to be a notable different between linear and affine conceptor but it doesn't seem to be the case. How do you explain this? What happens to the computed $C$ matrix when $\mu_c$ is used?

**Q2**: The seek for methods to steer transformers beyond CAA-fashioned steering vectors has been round for some time. Steering vectors have seen improvements like FVs[1] and BiPO[2], steering can be achieved through more sophisticated transformations like flow matching[3] and sparse autoencoder features[4], even when limited to affine transformation, as the authors have cited themselves, there is existing research[5] on piecewise linear projections for mean/covariance matching. It is not reasonable still compete against CAA only and it is important to know how well does conceptor steer LLMs compared to other steering methods. Do conceptor have fundamentally better outcomes?

**Q3**: While the AND-combined conceptor outperformed steering vectors in Figure 4, the gap between AND-combined and oracle conceptor is not significantly smaller than that between the added and oracle steering vectors. How does that prove that conceptors are better at representing composite logic?

**Q4**: What happens to $\mu_c$ in the residual steering function?

[1] Todd, Eric, et al. Function Vectors in Large Language Models. 2023. openreview.net, https://openreview.net/forum?id=AwyxtyMwaG.
[2] Cao, Yuanpu, et al. Personalized Steering of Large Language Models: Versatile Steering Vectors Through Bi-Directional Preference Optimization. 2024. openreview.net, https://openreview.net/forum?id=7qJFkuZdYo.
[3] Wang, Hanyu, et al. TruthFlow: Truthful LLM Generation via Representation Flow Correction. 2025. openreview.net, https://openreview.net/forum?id=7TDnfx5s14&noteId=5e01x1KGQu.
[4] Chalnev, Sviatoslav, et al. Improving Steering Vectors by Targeting Sparse Autoencoder Features. arXiv:2411.02193, arXiv, 21 Nov. 2024. arXiv.org, https://doi.org/10.48550/arXiv.2411.02193.
[5] Singh, Shashwat, et al. Representation Surgery: Theory and Practice of Affine Steering. 2024. openreview.net, https://openreview.net/forum?id=GwA4go0Mw4.

**Ethical Concerns:**

["NO or VERY MINOR ethics concerns only"]

**Final Justification:**

The authors provided additional clarification on their "automatic aperture tuning" process. However, this did not resolve the underlying technical concerns. It introduced inconsistencies with the appendix and reinforced the impression that the method is not robustly handling key phenomena like mean shift. Therefore, my major initial concerns remain largely unresolved. The authors still didn’t adequately justify why conceptors are a suitable or superior choice compared to other methods. The proposed solution to address criticism of the paper's main claims (interpretability, compositionality) is to remove or refine their claimed contributions, which would significantly weaken the paper's contribution in overall. In total, the rebuttal and discussion, though the effort of which is appreciated, does not improve the quality of the work significantly and thus I decided to keep my original ratings.

**Limitations:**

The authors claimed that there is detailed discussion of limitations in the discussion section but I don't find a discussion section. But I do see limitations being discussed in the conclusion section.

**Quality:**

3

**Strengths And Weaknesses:**

Strength:
+ The idea of applying conceptor onto transformers for activation steering is novel yet straightforward.
+ It is nice that the authors even included comparison with LoRA in the appendix.

Weekness:
+ **Unjustified claims**:
    + The authors claimed that the proposed steering method performs "interpretable control" and is "a powerful tool for the future of LLM control and interpretability", but there is no attempt to interpret the steering or the models being steering at all. Throughout the paper, only once on L81 did they ever talked about interpretability again and it seems that they were just equating linearity to interpretability.
    + The authors regarded "most activation addition work" as "primarily empirical without strong justification", but the authors themselves are not making strong justification either. The so-called proofs are mostly taken from the [2] with minor adaptions and the authors are not very strict about the assumptions, wording and definitions. For instance as I will ask in the questions section, I have concerns about how the authors understand conceptor given the Figure 1 they showed.
    + The authors said conceptor "can modify activations along multiple directions simultaneously" and "represent more complex steering functions than an additive steering function" due to more parameters used. However, being linear, conceptor isn't known for complexity to begin with and additive steering vectors can also be applied on $D$ layers or heads, etc. to consume $D^2$ parameters and point to different directions.
+ **Lack of baselines". As I will ask in the questions section, there have been many improvements since vanilla steering vectors, some of which are even being cited by the authors. Then why don't the authors compare conceptor with them. It seems late to the party if the authors just want to repeat how insufficient and unreliable vanilla steering vectors can be. Given that they have made comparison with LoRA, I think they are capable of incorporating more baselines to make the paper more wholesome.
+ **Poor writing** in general and in specific to citations:
    + The authors habitually quoted from or referred to texts/terminologies from the reference papers they cited without due explanations or adaptions. For instance, section 2.1 is more or less copied from section 2 of [1] except for dropping the binary classification notion of concepts. What's even worse is that due to the drop, there is no sentences left for defining what is a concept. The concept-encoding function being a function, indicates that each string has only one concept but then the authors again says there are concepts that "may be active" in the string. What is "may be"? What is "active"? Are they talking about substrings? The quoted sections was already not very easy to follow and the lack of efforts in rewriting only made it more incomprehensible. The authors also mentioned "guardedness" from [1] but even after reading the appendix, I have no clue what that means and I have to read [1] to know that it's defined on the linear non-separability of the binary concept states after concept erasure. The authors are replacing the "guardedness" requirement with their "regularized identity" requirement can claims optimality under the new requirement but didn't even make it clear what the former is let alone justifying why the latter is better.
    + The citations are inaccurate either. For instance [2] and [3] are both counted twice in the reference list, and when the authors quoted [4] in L21, they are actually referring [4] repeating the conclusions made in InstructEval.

[1] Singh, Shashwat, et al. Representation Surgery: Theory and Practice of Affine Steering. 2024. openreview.net, https://openreview.net/forum?id=GwA4go0Mw4.
[2] Jaeger, Herbert. Controlling Recurrent Neural Networks by Conceptors. arXiv:1403.3369, arXiv, 17 Nov. 2024. arXiv.org, https://doi.org/10.48550/arXiv.1403.3369.
[3] Jorgensen, Ole, et al. Improving Activation Steering in Language Models with Mean-Centring. arXiv:2312.03813, arXiv, 6 Dec. 2023. arXiv.org, https://doi.org/10.48550/arXiv.2312.03813.
[4] Chen, Banghao, et al. Unleashing the potential of prompt engineering in large language models: a comprehensive review. ArXiv, abs/2310.14735, arXiv, 2023

---

> ### Author Rebuttal · Authors · 2025-07-31
>
> ## Weaknesses
>
> *Concerning interpretability*:
> As we discussed in our response to reviewer `Fmbn` (Weakness 3), we rectified all unsupported claims regarding interpretability and instead point to conceptors as a potentially promising tool for interpretability, and an interesting subject for further mechanistic analysis - while making it clear that we provide no/limited insights into this in the present paper.
>
> *Concerning the complexity of steering functions*:
> Our claim refers to _functional_ expressiveness, not to raw parameter count, and we have made this more clear in our paper.
> - No matter how many layers or heads we steer with additive vectors, each update is still a *translation* in its local $D$-dimensional space; it cannot rescale, rotate, or mix coordinates within that layer.
> - Importantly, conceptors also demonstrate *input-dependent steering* because the impact of a conceptor depends on the current activation $h$, whereas additive vectors have a constant effect independent of the token context.
> - Finally, the affine conceptor defined in Section 2.4 is *strictly more expressive* than vector steering as it combines additive steering with the linear multiplicative conceptor matrix (for aperture $\alpha \rightarrow \infty$, affine conceptor steering reduces to additive steering).
>
> *Concerning the definition of concepts*:
> We appreciate the reviewer’s concern and agree that Section 2.1 should be made clearer. We are rewriting the theoretical framework section in our own words with less reliance on the work of Singh et al (2024). We further introduce an explicit, self-contained definition of concepts:
> > Let $\Sigma^{\ast}$ be the set of token sequences produced by the language model. A concept (or behavioral property) is a measurable function $c:\Sigma^\ast \longrightarrow [0,1]$, where $c(x)$ quantifies the degree to which sequence $x$ manifests the property. This function features gradedness (as opposed to binary concepts that are either present or not), where $c(x)=1$ means the property is fully present, $c(x)=0$ fully absent, and intermediate values reflect partial realization or uncertainty. We further allow for compositionality, meaning that multiple concepts/behaviors can be active simultaneously. For example, a reply can be truthful ($c_{\text{truth}}(x) \approx 1$) and friendly ($c_{\text{friend}}(x) \approx 1$). We define activeness of a conceptor (or behavioral property) $c$ in a text $x$ if $c(x) > \tau$ for a chosen threshold $\tau \in [0,1]$.
>
> *Concerning the definition of guardedness*:
> We apologize for the incorrect reference to the Appendix and the missing information about the concept of guardedness. We have rewritten the section to refer less strongly to the work by Singh et al. (2024). As reviewer 1517 pointed out, $\phi$-assisted steering functions and guardedness do not play a primary role in our theoretical framework and we therefore moved these concepts and references to the related work section. We have made sure to include a concise and correct definition of guardedness in this section.
>
> *Concerning citations*:
> We thank the reviewer for identifying these citation errors and apologize for the oversight. Should a problem remain with the reference to Chen et al (2023), we are happy to also resolve this, if the reviewer could kindly clarify the problem. We have now removed the duplicate entries in our bibliography.
>
> ## Question 1
>
> This is correct - classically, conceptors assume states to be zero-mean. As the reviewer points out, this does not hold for activations in LLMs (see our response to reviewer 1517's Q1 above). The extreme example nicely points out the shortcoming of linear conceptors. This is precisely why we derive affine conceptors, and we have made this motivation more clear in our paper. When $\mu_c$ is used, the conceptor $C$ will not pick up on the mean activation direction, as the conceptor will effectively be computed on the mean-centered activations and it will also be applied to mean-centered activations.
>
> However, we wish to clarify that a linear conceptor can still pick up meaningful directions in the outlined (extreme) example. With well-chosen aperture $\alpha$, the conceptor is not "dominated by the mean shift while ignoring the actual axis of variance".
>
> In the above example, if activations cluster around $v\pm\varepsilon\cdot v^\perp$, then the covariance matrix is given by covariance matrix $\hat \Sigma_c \propto \| v \|^2 v v^\top + \varepsilon^2 v_\perp v_\perp^\top$. In other words, $v$ will be an eigenvector of the with eigenvalue $s_v \approx \| v \|^2$, and $v^\perp$ will also be an eigenvector with eigenvalue $s_{v^\perp} \approx \varepsilon^2$. The aperture $\alpha$ of the conceptor $C(\tilde \Sigma_c)$ scales the eigenvalues of the conceptor as:
> $$\hat s_i = \frac{s_{i}}{s_{i} + \alpha^{-2}}$$
> Taking $\|v\| = 10$ and $\varepsilon = 0.1$ (such that $\|v\| = 100 \ \varepsilon$) we can get $\hat s_{v} \approx 1$ and $\hat s_{v^\perp} \approx 1$ if we choose $\alpha = 1000$. We kindly refer to our comment to reviewer `Xs77` (Weakness 1) for a description of our automatic aperture range mechanism which would lead us to (by definition) use apertures in the correct range in the above example.
>
> ## Question 2
>
> The reviewer correctly identifies a key challenge in this field—most recent activation steering methods introduce novel benchmarks with different evaluation protocols, models, and tasks, making direct comparison difficult. We initially focused on widely-cited methods (Todd et al. 2023, Rimsky et al. 2024) that have established evaluation frameworks.
>
> To directly address this concern, we implemented a comparison with function vectors (FVs, Todd et al. 2024), which allows for evaluation on identical tasks and conditions. We have added this comparison to our paper. We evaluated conceptor steering on the same 6 tasks used in their work under identical zero-shot conditions.
> For each model, we selected the central layer and optimized only the aperture hyperparameter (keeping $\beta=1.0$), evaluating performance across 20 random seeds. Our results demonstrate consistent improvements over function vector steering across all tested models, including the 70B parameter Llama-2 model:
>
> |                | Layer | Conceptor Accuracy  | FV (Todd et al.)    |     |
> | -------------- | ----- | ------------------- | ------------------- | --- |
> | GPT-J (6B)     | 15    | 73.1 &plusmn; 1.1 % | 57.5 &plusmn; 1.7 % |     |
> | GPT-NeoX (20B) | 15    | 82.1 &plusmn; 0.6 % | 57.1 &plusmn; 1.5 % |     |
> | Llama-2 (70B)  | 40    | 89.6 &plusmn; 1.1 % | 83.8 &plusmn; 0.7 % |     |
>
> In addition to being less performant than conceptor steering, FVs have several properties that make them computationally prohibitive and less desirable than conceptor steering and the additive steering baseline we used, which is why we did not consider them as a viable baseline. *First, FVs use more information.* They not only require the token embedding at a given layer $x^l \in \mathbb{R}^d$ but also the output of the attention layer from all individual heads $a_{l,j} \in \mathbb{R}^d$ for $j \in \{1, ..., J\}$ across multiple layers $l$ for all the examples in the task. This amounts to >500,000 attention head activations compared to only ~100 token embeddings required by conceptor steering and additive steering. *Second, FVs require more inference paths through Causal Mediation Analysis.* For a single task, this requires >100,000 inference paths, whereas conceptor/additive steering only requires one per example.
>
> We further evaluated conceptor steering using the Llama-2-7B model on the TruthfulQA benchmark to compare against the TruthFlow method by Wang et al. (2025), which was mentioned by the reviewer. Our results show that the performance of TruthFlow lies within the standard error of the performance of our conceptor steering method. We did not optimize the hyperparameters for the conceptor, we simply took the best hyperparameters from the `coordinate-other-AIs` multiple-choice task.
>
> | Method | MC1 | MC2 |
> | --- | --- | --- |
> | Baseline | 32.03 | 49.51 |
> | Conceptor | 33.49 $\pm$ 2.3 | 49.01 $\pm$ 2.2 |
> | TruthFlow | 34.47 | 51.82 |
>
> Unfortunately, Wang et al. (2025) [did not report statistical validations of their results](https://arc.net/l/quote/wgujbfyk), which makes it difficult to assess if the difference in performance is significant. Moreover, we highlight that the flow matching method proposed by Wang et al. (2025) involves multiple gradient steps and is significantly more expensive than our conceptor steering method. If the reviewer deems this a helpful baseline comparison, then we can include fully optimized results in our final paper for a fair comparison against TruthFlow.
>
> ## Question 3
>
> It is correct that our current analysis does not definitively isolate whether the superior performance stems from the compositional operation itself (the AND operation) versus the general effectiveness of conceptor representations.
>
> To address this concern, we measured the relative degradation in performance between the best performing layer when using combined methods versus optimal (oracle) performance. The results supports our original claims: conceptors demonstrate superior compositional efficiency on 2 out of 3 task pairs (English-French & Capitalize: -45.3% vs -52.0% degradation; Singular-Plural & Capitalize: -5.6% vs -13.3% degradation). The exception is English-French & Antonyms, which notably represents the task where compositional steering faces significant difficulties for both conceptor and activation approaches (as clearly shown in Figure 4, left).
>
> Given these findings, we propose to refine our claim to be more precise about our current evidence and distinguish relative degradation and absolute performance.
>
> ## Question 4
>
> We have updated the theoretical framework section to be more consistent and easier to read to address this question.

---

> > ### Comment · Reviewer_mWE6 · 2025-08-07
> >
> > I thank the authors for their attempt to fix the problems pointed out by me and also other reviewers. However, many of my concerns are still not lifted. 1) The authors showcased their "automatic aperture tuning" process which is different from the grid search reported in Appendix D, and the range of the aperture searched over there is highly disjoint with the one they suggested for my example. While for extremely large aperture, of course one can preserve the info in mean shift because the projection is dominated by identity but that is not usefully giving the "optimal" linear transformation anymore. Basically, I still think the optimization in Eq. 8 is too under-specified to account for mean shift on its own and one should not anticipate good results without the reintroduction of linear steering under the authors' current theoretical framework. 2) The authors restated their envisioned expressiveness of conceptors but as the authors have realized themselves, that is basically the common trait of all "input-dependent steering" which, if one wants, can go much more complex than conceptors while the authors cannot justify why conceptors is the best choice. Based on their new results against TruthFlow, conceptors are not as effective as this other "input-dependent steering" method (if the authors insist that TruthFlow lie within the standard deviation then conceptors are also not significantly outperforming the naive baseline), which adds to my concern. 3) The authors promised to remove or refine their claims like interpretability and compositionality, but in that case, the contribution of this work is highly reduced. In overall, I find my major concerns with this paper still persisting and hence would like to maintain my current ratings.

---

### Official Review · Reviewer_Xs77 · 2025-07-02

**Clarity:** 3
**Significance:** 3
**Originality:** 3
**Rating:** 3
**Confidence:** 1

**Summary:**

The paper proposes Conceptor-based compositional affine activation steering, a plug-in technique that edits an LLM’s hidden states during inference without touching its weights.
Instead of adding a single shift vector, it projects activations through a covariance-derived “soft mask” $C$ or residual form
$I+C$, which can be logically combined via AND/OR/NOT to coordinate multiple concepts.
This yields distribution-aware, precisely tunable control and beats additive steering—and even matches or surpasses full-rank LoRA—on several function and safety benchmarks with negligible compute overhead.
The approach, however, incurs extra hyper-parameter tuning and quadratic parameter growth, and its scalability to giant models or open-ended generation remains an open question.

**Questions:**

Same as Weakness

**Ethical Concerns:**

["NO or VERY MINOR ethics concerns only"]

**Final Justification:**

I appreciate the author's response and appreciate their effort. However, I am still concerned about the instability of the model in real life due to hyperparameters.

**Limitations:**

yes

**Quality:**

3

**Strengths And Weaknesses:**

Strengths
1. This paper provide solid theoretical foundation with a closed-form solution. The authors recast optimal linear/affine steering as a regularized minimization problem whose unique closed-form solution is the Conceptor matrix, unifying Conceptor theory with activation steering.

2. By using the covariance matrix, a Conceptor scales each principal component smoothly between 0 and 1. Activations already aligned with the concept are left almost untouched, while distant ones are pulled in—reducing distribution shift compared with a single “shift vector.”

3. Conceptor can achive better empirical performance at lower cost. On GPT-J 6B, GPT-NeoX 20B and other models, Conceptor steering beats additive vectors—and even matches or surpasses full-rank LoRA—while requiring no gradient updates and far less compute.


Weakness and Limitations
1. The method has higher computation and tuning overhead.
It must estimate a covariance matrix and tune aperture ε plus strength φ—more work than choosing a single shift vector.

 2. A Conceptor contains $D^2$ parameters (vs. $D$ for an additive vector), so it needs more positive examples to avoid noise or overfitting.

3. This paper current has limited scope of evaluation. Results are shown only on mid-scale models and short, synthetic tasks; performance on larger LLMs, long-form dialogue, or multilingual settings remains untested.

---

> ### Author Rebuttal · Authors · 2025-07-31
>
> ## Weakness 1
>
> As we argue in the conclusion of our paper, conceptor-based steering function occupy a middle ground between simple steering vectors and fine-tuning of language models. Conceptors are more costly than steering vectors but they are also far more performant. Most importantly, conceptors are far more efficient than fine-tuning while also showing better performance (see Figure 3 and Appendix D.2).
>
> The cost of computing steering vectors is actually comparable to that of conceptors. Given a set of concept-positive activation vectors arranged in a matrix $X \in \mathbb{R}^{N \times D}$, steering vectors are computed as the average activation whereas conceptors are calculated using the closed form solution in Eq. 9. The cost of calculating the steering function from $X$ is negligible in comparison to the forward passes of the LLM to get the activation vectors.
>
> The reviewer also raises the problem of having an extra hyperparameter, the aperture $\alpha$, that requires tuning. Both steering vectors and conceptors require tuning the steering strength $\beta$.
>
> For our experiments, we have developed an **automatic aperture tuning mechanism** that greatly simplifies the tuning of the aperture value based on the singular value spectrum of the covariance matrix $\tilde \Sigma_c$. We describe this mechanism below, and added a new section in the Appendix with full details on its implementation and results.
>
> ### Automatic aperture tuning
> Following Proposition 1, the conceptor matrix has a closed-form solution depending on the covariance matrix and the aperture: $C(\tilde \Sigma_c, \alpha) = \tilde \Sigma_c \left( \tilde \Sigma_c + \alpha^{-2} I \right)^{-1}$. If $\tilde \Sigma_c$ has singular value decomposition $\tilde \Sigma_c = \Sigma_{i=1}^D s_i u_i u_i^\top$, then the conceptor $C$ has the same eigenvectors but transformed eigenvalues, according to $\hat s_i = \frac{s_{i}}{s_{i} + \alpha^{-2}}$, where $s_{i}$ are the eigenvalues of $\tilde \Sigma_c$ and $\hat s_i \in [0,1)$ are the eigenvalues of $C$.
> - We determine a lower and upper bound of aperture values based on the singular value spectrum of the conceptor. The upper and lower bound of values are the apertures at which the lowest and highest singular values reach a threshold of $1 - \varepsilon$ (where $\varepsilon$ is machine precision in float32).
> - We estimate machine precision by $\epsilon \times |s| * 25 * \sqrt{max(s)}$. Additionally, we enforce an upper bound of machine precision to lay at 0.9999, since the maximum singular values of the covariance matrices explode in later layers, pushing our formula past the upper bound of singular values.
> - Using $\hat s_i = \frac{s_{i}}{s_{i} + \alpha^{-2}}$, we get $min\_aperture^{-2}=max(s) \times \frac{1- \text{threshold}}{\text{threshold}}$ and $max\_aperture^{-2}=min(s) \times \frac{1- \text{threshold}}{\text{threshold}}$.
> - We have observed empirically that this range of aperture values is the most expressive. In/Decreasing apertures beyond these bounds will not yield significantly different steering performance as the conceptors' singular value spectra do not change significantly beyond this range.
> - We note that an existing conceptor $C(\alpha_1)$ with aperture $\alpha_1$ can be rescaled to another aperture value $\alpha_2$ with $C(\alpha_2) = C(\alpha_1) \left( C(\alpha_1) + (\frac{\alpha_1}{\alpha_2})^2 (I-C(\alpha_1)) \right)^{-1}$ as described in [Jaeger (2014)](https://arxiv.org/abs/1403.3369).
> - In summary, this procedure is cheap and directly reduces the search space for optimal apertures in the hyperparameter sweep.
> We further refer to our response to reviewer `Fmbn` (Weakness 1), where we outline a method for reducing the search space for optimal layers where conceptor steering is applied.
>
> ## Weakness 2
>
> We agree with the reviewer that this is a natural expectation to have - since conceptors have $D^2$ parameters, they may be expected to require more training samples than vectors which have only $D$ parameters. However, this is actually not the case as it is confusing the number of entries in the conceptor matrix with the number of free (fitted) parameters:
> - Although a conceptor matrix $C\in\mathbb{R}^{D\times D}$ *contains* $D^{2}$ entries, they are *not* independently fitted.  Equation (9) shows how $C$ is a deterministic shrinkage transform of the *second-moment* matrix $R$.
> - $R$ is symmetric PSD, hence described by its $D$ eigenvalues and their directions; no extra free parameters are introduced. When the number of samples $n$ is fewer than the activation dimension $n<D$, $\operatorname{rank}(R)\le n$, so the conceptor is actually *low-rank* in most of our applications where $n << 1000 < D$.
> - In terms of sample complexity, standard covariance-concentration bounds, e.g. $\lVert\hat R-R\rVert_{2}=O_p \bigl(\sqrt{D/n}\bigr)$, imply that $O(D)$ samples suffice for stable eigen-structure estimation— the same order as for a steering vector.
> - The Tikhonov term $\alpha^{2}I$ shrinks all eigenvalues into $[0,1)$, capping variance and preventing over-fitting even when $n\ll D$.
> So, despite its larger raw size, a conceptor’s *capacity* is effectively comparable to that of an additive steering vector, and its required number of positive examples is governed by covariance estimation, *not* by $D^{2}$. This matches our experiment, where test accuracy saturates for both methods while the conceptor attains the higher plateau.
>
> As a simple experiment to demonstrate this empirically, we take three of the function vector task with the largest dataset (`antonyms`, `capitalize`, `english-french`), and vary the number of "training examples" that we use to compute steering vectors and conceptors, and report the accuracy on un-seen examples. We chose the best hyperparameters for each task and steering method based on the results from our extensive grid search that produced Figure 2 in our paper.
> We note that every "sample" contains 10 in-context examples of the function, thus the effective number of training examples is ten times what is reported in the table below.
>
> | # Samples | Conceptor (test acc.) | Vector (test acc.) |
> | --- | --- | --- |
> | 1   | 0.0 ± 0.0 | 0.0 ± 0.0 |
> | 2   | 29.5 ± 15.1 | 17.6 ± 11.5 |
> | 4   | 54.9 ± 5.0 | 36.6 ± 2.7 |
> | 6   | 55.4 ± 4.7 | 39.2 ± 2.1 |
> | 8   | 55.9 ± 3.8 | 39.9 ± 1.6 |
> | 10  | 56.3 ± 2.7 | 39.4 ± 2.3 |
> | 20  | 57.8 ± 1.8 | 40.6 ± 1.1 |
> | 40  | 58.4 ± 1.8 | 40.6 ± 1.0 |
> | 60  | 59.0 ± 1.4 | 40.7 ± 0.7 |
> | 80  | 57.4 ± 4.3 | 40.8 ± 0.6 |
> | 100 | 58.8 ± 1.0 | 40.7 ± 0.6 |
>
> It can be seen that the accuracy saturates after ~20 samples for both conceptors and steering vectors. Conceptors reach their optimal accuracy with 60 samples and steering vectors with 80 samples. This shows that conceptors do not require more training samples for maximum performance, but they do reach significantly higher performance.
>
> ## Weakness 3
>
> We have demonstrated better-or-equal performance of conceptors on synthetic tasks using LLMs with up to 20B parameters. We also demonstrated compositionality of conceptors in multilingual settings by composing the English-French function with two other functions (see Figure 4).
>
> We further report results on open-ended language generation in Section 3.4. We believe that these experiments demonstrate the viability of conceptor-based steering for long-form dialogue, as models are evaluated on open-ended generation rather than simpler tasks such as function vectors and multiple-choice answers. We kindly refer the reviewer to Appendix D.6 for more details on the experimental setup for open-ended language generation. During this rebuttal period, we have further extended the range of datasets, models and model sizes for a comprehensive comparison of conceptors against contrastive activation addition - we present our results in our response to reviewer `Fmbn` (Weakness 1). We also present extended results on open-ended steering on Llama-2-7b-chat (the model used by Rimsky et al. (2024)) and on multiple datasets, for which we kindly refer you to our response to reviewer `1517` (Weakness 5). We will update our limited (and incomplete) results in Figure 5 with the comprehensive results presented in this rebuttal.
> We agree with the reviewer that results for larger-scale LLMs (beyond 20B parameters) would be beneficial to demonstrate the scalability of our proposed method. During this rebuttal, we have therefore added results for the Llama-2-70b model on function vector tasks - where conceptors strongly outperform the much more expensive function vectors by Todd et al (2024), see our response to reviewer `mWE6` (Question 2) and reviewer `1517` (Question 3). We further added results for Llama-2-7b and Llama-2-70b for complex behavior tasks, see our response to reviewer `Fmbn` (Weakness 1) and reviewer `1517` (Weakness 5).

---

> ### Comment · Reviewer_Xs77 · 2025-08-07
>
> I appreciate the author's response and appreciate their effort. However, I am still concerned about the instability of the model in real life due to hyperparameters. To strengthen the rebuttal, I recommend additions: (i) quantify the practical savings of the automatic‐aperture routine by reporting average GPU-hours and trial counts versus naïve grid search across several tasks, (ii) extend the evaluation to truly large models (e.g., Llama-2 13B/34B or Mixtral) and more realistic settings such as multi-turn dialogue or open-domain QA to demonstrate scalability. Given these concerns, I maintain the score.

---

### Official Review · Reviewer_Fmbn · 2025-07-02

**Clarity:** 3
**Significance:** 3
**Originality:** 3
**Rating:** 4
**Confidence:** 3

**Summary:**

This paper introduces a framework for controlling the behavior of large language models (LLMs) by integrating conceptor theory with activation steering. The authors propose conceptors, which are essentially soft projection matrices, to compress and guide sets of activation vectors, offering more precise control over the model's internal states. The authors derive optimal affine steering functions grounded in theory, show how conceptors enable linear transformations rather than simple additive shifts, and present Boolean operations (AND/OR/NOT) to combine steering objectives.

**Questions:**

1. How robust is conceptor-based steering under distribution shift, noise (i. e. test for robustness)?

2. Missing reference in related works: 1. Effectively Steer LLM To Follow Preference via Building Confident Directions 2. LLMSteer: Improving Long-Context LLM Inference by Steering Attention on Reused Contexts

**Ethical Concerns:**

["NO or VERY MINOR ethics concerns only"]

**Final Justification:**

I'd like to maintain my score as is it given that I believe it is beneficial to dissect the effect of different aperture values and residual steering application as an important ablation study. Therefore, I can't further increase my score to "accept".

**Limitations:**

yes

**Quality:**

3

**Strengths And Weaknesses:**

**Strengths**
1. Significance/Quality: Provides a mathematically derivation of linear and affine steering functions with closed-form solutions, generalizing prior additive approaches.
2. Quality: Authors compare against LoRA adapters, showing comparable or superior results with lower computational cost.
3. Quality: The authors have conducted experiments across multiple models (Transformers and SSMs).
4. Clarity: the paper is easy to follow.

**Weaknesses**

1. The proposed method requires computing covariance matrices and tuning additional hyperparameters (mentioned in L327). The scalability to very large LLMs remains unproven, in fact, most of the tested LLM are relatively small. The paper could benefit from a more detailed analysis of this trade-off.
2. No analysis (ablation study) of how aperture or residual steering contribute independently.
3. Despite claiming "interpretable control" (Abstract), no analysis/visualization demonstrates *how* conceptors alter internal mechanisms. It can be done by probing, attention visualization

---

> ### Author Rebuttal · Authors · 2025-07-30
>
> ## Weakness 1
>
> It is true that conceptors are more expensive to compute than vanilla steering vectors. However, as in our response to weakness 1 by reviewer `Xs77`, we argue that the cost of computing a conceptor is negligible in comparison to the number of forward passes through the model that other competing methods require - see Appendix D.2 for a comparison to LoRA finetuning, and our response to Question 2 by reviewer `mWE6` for a comparison to function vectors by Todd et al (2024) and flow matching by Wang et al (2025).
>
> Concerning the additional hyperparameter, we kindly refer the reviewer to our response to reviewer `Xs77` (Weakness 1) where we address this concern. In short, we propose a general method to determine relevant ranges for the aperture parameter and explain how a single conceptor can be efficiently rescaled to different aperture values.
>
> In addition to reducing the aperture parameter's sweep range, we also propose a conceptor-specific method for reducing the hyperparameter search space by restricting the number of layers that are sweeped. We have observed that across all our experiments, the conceptor steering method generally does not perform well in early layers. We found empirically that the best conceptor steering performance occurs in the 20% of layers after which the covariance matrix $\tilde \Sigma_c$ has reached maximal rank. The maximal rank is typically determined by the number of samples that are used to compute the covariance matrix. This further reduces the hyperparameter search space for conceptors greatly. We have added a new section in the Appendix with more details on this procedure.
>
> For the scalability to larger LLMs, we point out that the GPT-NeoX model has 20B parameters and, as part of this rebuttal, we also supply additional results for larger models. We show results on the function vector task for the Llama-2-70B model where *conceptors outperform function vectors* (a much stronger baseline than simple activation vectors, by Todd et al. (2024)), see our response to reviewer `mWE6` (Question 2) and include the full results table in our response to reviewer `1517` (Question 3).
>
> Below, we also add results for all complex behavior tasks from Rimsky et al (2024) on the llama-2-7b and llama-2-70b models. We further update our results for Mamba-2.8B which we have improved with our new aperture tuning recipe and more principled hyperparameter search. We include results for baseline (no steering), contrastive activation steering, conceptor steering, and contrastive conceptor steering (by combining the conceptor $C_+$ for matching behaviors and the conceptor $C_-$ for non-matching behavior and combining them with $C=C_+ \wedge \neg C_-$). Due to time contraints of this rebuttal period, we were unable to finish our results for contrastive conceptors on Llama-2-70b.
>
> These results show that *conceptors consistently match or outperform activation steering across tasks, models, and model sizes*. One anomaly is that conceptors are not good at steering for refusal (83% vs. 91% for activation steering), and we are currently looking into reasons for this. For convenience, we have added the standard deviation across tasks for this model size. More experiments are needed to investigate the steering performance of conceptors vs. activation vectors on larger models for complex behavioral steering.
>
> ### state-spaces/mamba-2.8b-hf
> | | coordinate-other-ais  | corrigible-neutral-HHH | refusal | myopic-reward | hallucination | sycophancy | Average |
> | --------------------- | ---------------------- | ------- | ------------- | ------------- | ---------- | ------- | ---- |
> | Baseline              | 48.0                   | 42.0    | 56.0          | 52.0          | 50.0       | 47.0    | 49.2 |
> | Activation            | 63.0                   | 62.0    | 55.0          | 67.0          | 55.0       | 53.0    | 59.2 |
> | Standard Conceptor    | 78.0                   | 73.0    | 56.0          | 62.0          | 72.0       | 59.0    | 66.7 |
> | Contrastive Conceptor | 79.0                   | 75.0    | 62.0          | 72.0          | 66.0       | 56.0    | 68.3 |
> ### meta-llama/Llama-2-7b-chat-hf
> | | coordinate-other-ais  | corrigible-neutral-HHH | refusal | myopic-reward | hallucination | sycophancy | Average |
> | --------------------- | ---------------------- | ------- | ------------- | ------------- | ---------- | ------- | ---- |
> | Baseline              | 45.0                   | 46.0    | 54.0          | 51.0          | 45.0       | 44.0    | 47.5 |
> | Activation            | 77.0                   | 57.0    | 68.0          | 89.0          | 72.0       | 68.0    | 71.8 |
> | Contrastive Conceptor | 93.0                   | 87.0    | 66.0          | 85.0          | 93.0       | 67.0    | 81.8 |
> | Standard Conceptor    | 89.0                   | 98.0    | 66.0          | 91.0          | 99.0       | 69.0    | 85.3 |
> ### meta-llama/Llama-2-70b-chat-hf
> |                    | coordinate-other-ais | corrigible-neutral-HHH | refusal | myopic-reward | hallucination | sycophancy | Average |
> | ------------------ | -------------------- | ---------------------- | ------- | ------------- | ------------- | ---------- | ------- |
> | Baseline           | 52.0                 | 16.0                   | 30.0    | 48.0          | 57.0          | 13.0       | 36.0    |
> | Activation         | 93.0                 | 95.0                   | 91.0    | 100.0         | 91.0          | 93.0       | 93.8 $\pm$ 3.1    |
> | Standard Conceptor | 91.0                 | 98.0                   | 83.0    | 100.0         | 93.0          | 96.0       | 93.5 $\pm$ 5.6    |
>
> ## Weakness 2
>
> We agree with the reviewer that a more detailed ablation would be beneficial to analyze the isolated effect of different aperture values and residual steering application. Unfortunately, we have not had the time to implement this yet. If possible, we will consider including this in the final version of the paper - during the rebuttal we have focused on points that were more urgently raised by multiple reviewers (e.g., using larger models and comparing to more baselines from the literature).
> ## Weakness 3
>
> We enthusiastically agree with the reviewer that a more detailed mechanistic analysis of conceptor steering would be most interesting and important. We hoped to be able to provide this, but we have to accept that this is out of scope for the present paper. We apologize for the over-claiming in our abstract. We will rectify all unsupported claims and point to conceptors as a promising tool for interpretability, and an interesting subject for further mechanistic analysis and make it clear that we provide no/limited insights into this.
> ## Question 1
>
> This, too, is a very interesting question and we would be more than happy to explore this in the future. We have provided more results on the effect of increased number of training examples in our response to reviewer `Xs77` (Weakness 2) which may be somewhat relevant to the question about robustness to noise. Further, more targeted, experiments on robustness to noise and distribution shifts are sadly out of scope for us now, and we leave this open as exciting follow-up work.
> ## Question 2
>
> We thank the reviewer for pointing out additional references that are relevant to our work.
>
> We were not aware of the first paper by Song et al (2025) which is concurrent to our work, but we are happy to include it in our paper as an example of how steering can be applied for personalization in multi-user settings.
>
> The second paper by Gu et al (2024) on LLMSteer seems only indirectly relevant to our work as they are doing *attention steering* rather than *activation* steering. Rather than steering the model towards specific behaviors or concepts, as done in our line of research, attention steering guides the attention weights of an LLM towards specific parts of the input text. The goal of attention steering seems to be improving generation quality and long-context understanding, whereas the goal of activation steering is to control the behavior of LLMs more generally. However, we would be happy to include references to attention steering if the reviewer believes this will improve our paper.

---

> > ### Comment · Reviewer_Fmbn · 2025-08-05
> > **reply**
> >
> > Thanks authors for the detailed reply. I'd like to maintain my score as is it given that I believe it is beneficial to dissect the effect of different aperture values and residual steering application as an important ablation. Therefore, I can't further increase my score.

---

### Official Review · Reviewer_1517 · 2025-07-05

**Clarity:** 2
**Significance:** 2
**Originality:** 2
**Rating:** 3
**Confidence:** 4

**Summary:**

This paper applies conceptor theory, originally developed for RNNs, to construct linear/affine transforms for LLM activation steering. The proposed method is evaluated on composite tasks, behavior-based multiple-choice questions, and open-ended generation tasks. Results demonstrate that the conceptor-based steering approach surpasses the additive baseline—specifically, Function Vectors—on synthetic benchmarks. These findings suggest that conceptor-guided steering may serve as a contribution to the existing literature on activation steering techniques.

**Questions:**

1. Could the authors further explain the assertion that *"activations already residing within the conceptor's region experience minimal change, whereas activations outside this region undergo more substantial shifts"*? This claim is central to the proposed method, but its theoretical basis remains unclear.
2. Could the authors clarify the rationale for applying the expectation operation across all concepts $c$, or elaborate on the procedure if this is not the case?
3. Was there a specific reason for excluding Llama models, given their popularity and relevance in the compared baseline?

**Ethical Concerns:**

["NO or VERY MINOR ethics concerns only"]

**Limitations:**

Yes

**Quality:**

2

**Strengths And Weaknesses:**

Strengths
1. The study employs conceptor theories as a theoretical framework to systematically contrast existing approaches and develop combination techniques.
2. The reported results demonstrate consistent performance improvements over additive Function Vectors.

Weaknesses
1. The preliminaries largely mirror those of Singh et al. (2024). If the core contribution focuses on unassisted systems, the inclusion of assisted steering functions in this section is unnecessary—and omitting them would allow for a more concise presentation. Instead, this space could be more effectively utilized to elaborate on the concepts, offering greater theoretical or methodological clarity where necessary, as we explore in the questions below.
2. The paper's justification for conceptor linear steering remains unclear, particularly in contrast to additive steering (the baseline), which employs intuitive contrastive vector operations.
  - While the multiplicative operation $C\mathbf{H}_c$ could plausibly serve purposes like dimensionality reduction or feature smoothing, the claim that a single linear transformation matrix can *"minimize changes to representations already aligned with concept $c$ while effectively steering others toward $c$"* demands further theoretical or empirical support.
  - Similar concerns apply for affine steering. A more rigorous discussion would strengthen the argument.
3. The expectation operation in the loss functions appears to require that $C\mathbf{H}_c$ approximate $\mathbf{H}_c$ across all concepts $c$, which lacks a clear explanation.
  - Assuming that taking the expectation over diverse $c$ can aid in learning transferable features, this results in a matrix $C$ that implicitly encodes shared conceptual relationships. However, the evaluation framework is primarily focused on understanding the inherent properties of individual concepts.
  - Furthermore, if both \( C \) and \( C' \) are derived from a set of concepts, their composition might introduce theoretical inconsistencies.
4. The popular Llama series of models, as evaluated in Todd et al. (baseline), is missing.
5. The final experiment is notably incomplete—the paper explicitly acknowledges this, stating that key hyperparameter tuning was restricted to a <50% subset of layers and deferring a full analysis to the camera-ready version. This is quite unconventional in a research submission.
  - Furthermore, the concluding statement about advancing the field is explicitly made conditional on the results of these future experiments ("Should conceptor-steered open generation match the performance..."), which is also unusual and weakens the impact of the findings in the submitted manuscript.
  - Giving the paper the benefit of the doubt, we would appreciate if the authors could clarify in their rebuttal to what extent the current, presented experimental results (even with the acknowledged limitations) are considered self-contained and sufficient to support the paper's claims.

---

> ### Author Rebuttal · Authors · 2025-07-30
>
> ## Weakness 1
>
> Our core contribution is indeed to present the first theoretical framework for un-assisted activation steering functions. We have made this more clear in our manuscript and compressed the introduction of $\phi$-assisted steering functions which we agree is only marginally relevant given our aim (we moved the differentiation between our work and that of Singh et al. (2024) to the related work section).
>
> ## Question 1 & Weakness 2
>
> The claim that "activations already residing within the conceptor's region experience minimal change, whereas activations outside this region undergo more substantial shifts" stems directly from the mathematical formulation of conceptors and their eigenvalue structure.
>
> From Proposition 1, the conceptor matrix has the closed-form solution where $\tilde \Sigma_c$ is the second-moment matrix or covariance with the singular value decomposition  $\tilde \Sigma_c = \Sigma_{i=1}^D s_i u_i u_i^\top$. The conceptor $C$ has the same eigenvectors but transformed eigenvalues:
> $\hat s_i = \frac{s_{i}}{s_{i} + \alpha^{-2}}$
> where $s_{i}$ are the singular values of $\tilde \Sigma_c$ and $\hat s_i \in [0,1)$ are the eigenvalues of $C$.
>
> The eigenvectors $u_i$ are principal directions of the point cloud of training data points, and each $s_i$ equals the sample variance in that direction. Large variance in direction $u_i$ pushes the corresponding $\hat s_i$ close to 1, while small variance (or little support in the tranining data) gives $\hat s_i \approx 0$.
>
> The conceptor acts as a regularized identity map that adaptively scales activations along different principal
> directions. For activations $\mathbf{h}$ that align with high-variance directions of the concept (large
> $s_{c,i}$), the corresponding eigenvalues $s_i$ approach 1, meaning $C\mathbf{h} \approx \mathbf{h}$ (minimal
> change). Conversely, for activations along low-variance directions (small $s_{c,i}$), the eigenvalues $s_i$
> approach 0, causing substantial attenuation. This creates the ellipsoidal "conceptor region" where activations that lie within the high-variance subspace of the target concept experience preservation, while those outside undergo more significant transformation toward the concept's characteristic covariance structure.
>
> It is clear that affine conceptor steering is more expressive than additive steering because as $\alpha \rightarrow \infty$, this reduces to additive steering. Our motivation to introduce affine conceptors stems from the assumption of linear conceptors that samples are centered around the origin (the above-mentioned characteristic ellipsoid is always centered around the origin). This assumption clearly does not hold for activations of LLMs - otherwise additive steering vectors would not work.
>
> To validate that different functions lead to different mean activations, we performed statistical comparisons of mean activation patterns extracted at the colon token `:` during function vector tasks (see Appendix D.1). For each function type $f_i$ (e.g., antonyms, capitalization), we collected activation vectors $\mathbf{x}_{f_i}^{(j)} \in \mathbb{R}^d$ from $n=100$ examples.
> We performed one-sample t-tests against the null hypothesis that mean activations equal zero. We report the fraction of dimensions with $p < 0.05$ to quantify the extent to which function-specific representations deviate from the origin in activation space:
> ```
> Euclidean distance (antonyms   | zero):              73.641
> Euclidean distance (capitalize | zero):              74.112
> Mean absolute t-statistic (antonyms   | zero):       11.992
> Mean absolute t-statistic (capitalize | zero):       11.642
> Fraction significant p < 0.05 (antonyms   | zero):   89.58%
> Fraction significant p < 0.05 (capitalize | zero):   89.06%
> ```
>
> Interestingly, the difference in mean activations between the two functions is far less significant (euclidean distance 7.76, mean absolute t-statistic 1.19, fraction of significantly different dimension is 18.9%). These results can naturally also be used as an argument for the limitations of steering based on mean activation vectors.
>
> ## Question 2 & Weakness 3
> We apologize for the mistake in Eq. 8, the expectation is taken over the set of activations for a given concept $c$, i.e., $x \in \{ \mathbf{H}(s) \text{ where } \phi(s)=c \}$. The steering function $f_c(\mathbf{H}(s))$ is specific to one concept $c$ and therefore the conceptor $C$ is concept-specific and there is no averaging or expectation over all concepts $c$.
>
> We corrected this in our manuscript. We trust that the reviewer will now agree that there are no theoretical inconsistencies as conceptors are based only on individual concepts and there are no shared conceptual relationships. We further note that after fixing the typo in Eq. 8, our evaluation framework is entirely consistent with our theoretical framework.
>
> ## Question 3 & Weakness 4
> While we initially chose the Qwen series due to their increasing adoption and strong performance characteristics, we recognize that Llama models remain highly relevant for comparison purposes. We have added experiments on Llama-2-70B. We tested conceptor steering on the function steering experiment to complete our Figure 2 so that it can be visually compared with Figure 4 from Todd et al (2024).
>
> | Task \ Layer    | 0    | 10   | 20   | 30   | 40   | 50   | 60   | 70   | 79   |
> | --------------- | ---- | ---- | ---- | ---- | ---- | ---- | ---- | ---- | ---- |
> | antonyms        | 36.2 | 76.0 | 94.6 | 91.7 | 98.6 | 87.0 | 96.2 | 98.5 | 96.7 |
> | capitalize      | 39.4 | 78.1 | 86.6 | 95.3 | 99.4 | 97.8 | 97.9 | 99.9 | 99.8 |
> | country-capital | 59.5 | 77.5 | 80.1 | 84.1 | 74.8 | 68.3 | 66.3 | 80.1 | 79.6 |
> | english-french  | 25.7 | 73.2 | 93.6 | 92.3 | 94.7 | 91.5 | 96.5 | 90.4 | 93.7 |
> | present-past    | 8.6  | 60.3 | 82.0 | 97.2 | 85.5 | 48.1 | 47.6 | 60.4 | 42.2 |
> | singular-plural | 48.3 | 82.4 | 96.5 | 95.0 | 89.5 | 80.9 | 94.7 | 99.5 | 99.7 |
>
> These results demonstrate conceptor steering's effectiveness on Llama-2-70B across different layers. For a quantitative comparison showing that conceptors significantly outperform function vectors (a much stronger baseline than activation addition) on Llama-2-70B and other models, we kindly refer the reviewer to our response to Reviewer `mWE6` (Question 2).
>
> We further include results for complex behaviors (Section 3.4) for the Llama-2-7b and Llama-2-70b models in our response to reviewer `Fmbn` (Weakness 1).
>
> ## Weakness 5
>
> We apologize for the incompleteness of our final experiment section, we only included the full results in the supplementary materials (D.6). We re-ran all experiments at the last minute and did not get all experimental results in time for the paper submission. We apologize for the inconvenience and unconventional handling of this.
> The full results for the open-ended generation (Figure 5b) are included in the supplementary materials (D.6), and we paste the average performance for the best layer per steering method below. We note that, as described in Appendix D.6, we improved our scoring method to be a composite score of behavior and coherence, following [1].
> The results confirms that conceptors outperform contrastive vector steering by Rimsky et al. (2024). We have updated the results in our paper accordingly, and it is now entirely self-contained and sufficient to support our claims.
> ### Full results for Qwen-2.5-1.5B-Instruct:
>
> | Method             | Optimal layer | Avg. Performance | 95% CI      |
> | ------------------ | ------------- | ---------------- | ----------- |
> |  Additive steering | 17            | 20.0             | 3.2 - 30.7  |
> | Conceptor steering | 6             | 31.5             | 14.6 - 48.5 |
>
> ### Originally reported (incomplete) results for Qwen-2.5-1.5B-Instruct:
>
> | Method             | Layer | Avg. Performance |
> | ------------------ | ----- | ---------------- |
> | Additive steering  | 15    | 19.3             |
> | Conceptor steering | 15    | 8.08             |
>
> ### Open-ended steering performance on Llama-2-7b-chat
>
> Furthermore, as requested, we evaluated the steering performance on Llama-2-7B-Chat. Results represent the best improvement (across all layers) of the BCS (behavior-coherence score) in the steered setting with respect to the unsteered setting, and we include the 95% confidence interval. We attribute the high breadth of the 95% CI to our limited grid search (3 aperture values, 4 beta values) and the limited size of the test set at each layer (20 examples). Nonetheless, we think these results are directionally representative and serve to address the reviewer's question, while we work on larger hyperparameter sweeps and testing on more samples.
>
> | Steering method       | coordinate        | corrigible         | hallucination   | myopic            | sycophancy        |
> |-----------------------|-------------------|--------------------|-----------------|-------------------|-------------------|
> | Vector steering       | 7.20 ± 9.00       | 6.45 ± 11.53       | 7.75 ± 8.10     | 9.85 ± 11.48      | 6.38 ± 7.99       |
> | Conceptor             | **14.35 ± 13.62** | **12.475 ± 13.30** | 0.625 ± 4.87    | -1.525 ± 16.82    | **10.70 ± 11.64** |
> | Contrastive conceptor | 13.475 ± 10.76    | 11.60 ± 5.89       | **8.50 ± 8.03** | **17.03 ± 13.42** | 5.85 ± 13.14      |
>
> ## Summary
>
> We have addressed all weaknesses and questions by the reviewer and would like to thank them again for their keen eye and constructive criticism. We ensure that the clarify of our work has improved significantly through the points raised by the reviewer and we further hope that the reviewer finds it appropriate to raise their scores for the quality, significance and originality of our work.
>
> ## References
> - [1]: Samuel Soo, Chen Guang, Wesley Teng, Chandrasekaran Balaganesh, Tan Guoxian, and Yan Ming. Interpretable steering of large language models with feature guided activation additions, 2025. URL https://arxiv.org/abs/2501.09929.

---

> > ### Comment · Reviewer_1517 · 2025-08-04
> >
> > Your new results does not support your claim that conceptors significantly outperform function vectors on Llama-2-70B. In fact, conceptors and function vectors achieve the same best result in differenct layers. In my opinion, the highest accuracy among all layers is a more meaningful measure than the average accuracy across layers. Additionally, the data in Figure 2 of your paper is inconsistent with the data in Figure 4 on GPT-J and GPT-NeoX.

---

> ### Author Response · Authors · 2025-08-04
> **Conceptors do outperform function vectors on Llama-2-70B in a fair comparison**
>
> We refer to the table in our response to reviewer `mWE6` that show the performance of conceptor steering at a **single layer** and *averaged across all tasks* is $89.6 \pm 1.1$% whereas function vectors show $83.8 \pm 0.7$% at a single layer averaged across all tasks. We took the only numerical results reported by Todd et al (2024) in their Table 2 which chose layer 26 for llama-2-70b which is near-optimal (see their Figure 4). We chose layer 40 for conceptor steering because that seemed to be a good choice based on very preliminary experiments during this short rebuttal period. Thus, in a fair head-to-head comparison, conceptors outperform function vectors by 7% (relative).
>
> Figure 2 in our paper compares conceptors against additive steering vectors, not against function vectors. As we argued in our rebuttal, function vectors are significantly more expensive (>1000x) to compute than conceptors (see our response to reviewer `mWE6`). Therefore we present additive activation vectors as a more relevant baseline for comparison. However, we have already contacted the authors of the function vectors paper for their data for their Figure 4 so that we can add their results into our paper, but so far we have not received a response.

---

### Author Response · Authors · 2025-08-05
**A Friendly Reminder**

Dear Reviewers,

We sincerely thank you for the time and effort you dedicated to reviewing our work. We have carefully addressed each concern with detailed responses. Since the reviewer-author discussion period is closing in less than two days, we would appreciate it if you could check our response and see whether your concern has been addressed. If so, we kindly hope that you could consider updating the evaluation of the paper. We are pleased to answer your further questions.

-- The authors

---

### Note · Authors · 2025-08-14

We sincerely thank the reviewers for their feedback and have comprehensively addressed their concerns with extensive new experiments and clarifications. We are disappointed that reviewers cited new concerns not emphasized in their initial reviews and responded so late that we were unable to react to their responses.

We addressed the following reviewer concerns:
- Results for bigger models (Llama-2-70B) showing that conceptors still outperform other methods - `1517`
- Results for Llama-2 (7B and 70B) on all experiments - `1517`, `Xs77`, `Fmbn`
- Automatic aperture tuning mechanism to simplify hyperparameter search - `1517`, `Fmbn`, `Xs77`
- Analysis showing that conceptors are not more prone to overfitting and do not require more data than baselines - `Xs77`
- Direct comparison to stronger baselines (FV, TruthFlow) - `Xs77`, `mWE6`

Discussion summary:
- `1517`: we **addressed all concerns with concrete evidence**. The reviewer acknowledged but maintained the score of 3 citing concerns about "highest vs. average accuracy" which seems to be a **misunderstanding of our results**.
- `Fmbn`: we addressed the concerns about cost, hyperparameters, and scalability and offered to include more detailed ablation experiments. The reviewer **responded on the last day** insisting that ablations are necessary although **this was not marked as especially critical initially**.
- `Xs77`: we demonstrated that conceptors are cost-effective, don't require more samples, and scale to 70B models. The reviewer **raised entirely new concerns past the author response period** about even larger models and "multi-turn dialogue" (never mentioned initially).
- `mWE6`: we removed interpretability claims and added extensive baselines (FVs, TruthFlow). The reviewer **responded after the author response period**, introducing new criticisms about "under-specified optimization" and questioning why conceptors are "best choice" - **moving goalposts beyond original concerns**.

**Procedural Concerns**: reviewers responded very late (2/4 after the discussion period ended) and raised **new requirements post-rebuttal** not emphasized initially. The excessive use of "borderline" scores contradicts reviewer guidelines.

Given the late reviewer replies during this discussion period, we would be happy about an acceptance conditional on (1) further ablation studies (reviewer `Fmbn`), and (2) quantified practical savings of automated aperture tuning (reviewer `Xs77`).

---

### Decision · Program_Chairs · 2025-09-17

**Decision:**

Reject

**Comment:**

Reviewers appreciated the novel approach to affine activation steering, the principled theoretical foundation and empirical results, particularly the use of a “soft mask” to guide activations for enhanced LLM interpretability and control.

However, several critical concerns remain. Comparisons to advanced steering methods are lacking, scalability to larger models is unclear, and hyperparameter tuning is complex. Key theoretical claims on interpretability and compositionality are insufficiently substantiated. These unresolved issues prevent the paper from being accepted at its current form, though the work shows promise with further development and evaluation.